# Berbamine Reduces Chloroquine-Induced Itch in Mice through Inhibition of MrgprX1

**DOI:** 10.3390/ijms232214321

**Published:** 2022-11-18

**Authors:** Kunhi Ryu, Yunkyung Heo, Yechan Lee, Hyejin Jeon, Wan Namkung

**Affiliations:** College of Pharmacy and Yonsei Institute of Pharmaceutical Sciences, Yonsei University, 85 Songdogwahak-ro, Yeonsu-gu, Incheon 21983, Republic of Korea

**Keywords:** Chloroquine, MrgprX1, MrgprA3, berbamine, antagonist, itch

## Abstract

Chloroquine (CQ) is an antimalaria drug that has been widely used for decades. However, CQ-induced pruritus remains one of the major obstacles in CQ treatment for uncomplicated malaria. Recent studies have revealed that MrgprX1 plays an essential role in CQ-induced itch. To date, a few MrgprX1 antagonists have been discovered, but they are clinically unavailable or lack selectivity. Here, a cell-based high-throughput screening was performed to identify novel antagonists of MrgprX1, and the screening of 2543 compounds revealed two novel MrgprX1 inhibitors, berbamine and closantel. Notably, berbamine potently inhibited CQ-mediated MrgprX1 activation (IC_50_ = 1.6 μM) but did not alter the activity of other pruritogenic GPCRs. In addition, berbamine suppressed the CQ-mediated phosphorylation of ERK1/2. Interestingly, CQ-induced pruritus was significantly reduced by berbamine in a dose-dependent manner, but berbamine had no effect on histamine-induced, protease-activated receptors 2-activating peptide-induced, and deoxycholic acid-induced itch in mice. These results suggest that berbamine is a novel, potent, and selective antagonist of MrgprX1 and may be a potential drug candidate for the development of therapeutic agents to treat CQ-induced pruritus.

## 1. Introduction

Malaria is a life-threatening worldwide disease with concerning morbidity and mortality that caused approximately 627,000 estimated deaths in 2020 [1]. It is estimated that there were approximately 241 million cases of malaria in 85 malaria-endemic countries worldwide in 2020, with the African region accounting for 95% of an estimated 228 million cases in 2020 [1]. One of the main challenges in the elimination and control of malaria is the emergence of drug resistance [2]. Chloroquine (CQ) is a medication to prevent and treat malaria that has been used for more than 70 years. However, CQ-induced pruritus has been an obstacle to its clinical usage. Regardless of the rout of administration, CQ is known to cause a stinging or biting sensation several hours after administration [3]. CQ-induced pruritus occurs in 50% of Caucasians and dark-skinned Africans and African albinos, and the severity of pruritus varies from tolerable to intolerable in susceptible individuals [3]. CQ-induced pruritus has been an obstacle leading poor compliance in patients. Up to 30% of African patients are reluctant to have further CQ treatment due to the intolerable itch [4]. This poor compliance may develop CQ-resistant *Plasmodium falciparum* and accelerate the spread of malaria.

Since CQ-induced itch occurs after the first dose of CQ and cannot be treated with antihistamines, this suggests that the CQ-induced itch is not an allergic reaction and is a histamine-independent pathway [5]. Several G protein-coupled receptors (GPCRs) have been reported to play pivotal role in generating histamine-independent pruritus, such as Mas-related G Protein-coupled receptor (Mrgprs), protease-activated receptors (PARs), and Takeda G protein-coupled receptor 5 (TGR5) [6,7,8,9,10]. Some Mrgpr family members are distinctively expressed in small-diameter dorsal root ganglion (DRG) neurons and trigeminal ganglia neurons and have not been discovered in the central nervous system and other parts of body [11,12]. Mrgprs play a pivotal role in somatosensation, including itch and pain [7,13], and recently, it has been reported that MrgprX1 functions as a receptor for CQ and plays an essential role in CQ-induced pruritus, such as in the activation of small DRG neurons [7]. To date, a few antagonists have been reported to inhibit MrgprX1 or MrgprA3, a mouse ortholog of human MrgprX1, such as crotamiton and JHU739 [14,15]. However, crotamiton has low potency (IC_50_ = 326 μM) and low selectivity, and JHU739 inhibits the activation of MrgprX1 by BAM8-22, but the effect on MrgprX1 activation by CQ is unknown [14,15].

In the present study, we conducted a cell-based high-throughput screening and identified berbamine as novel potent and selective antagonist of MrgprX1. We further investigated the effect of berbamine on the CQ-induced activation of MrgprA3 and CQ-induced itch in mice.

## 2. Results

### 2.1. Berbamine, a Novel Antagonist of MrgprX1

To identify novel antagonists of MrgprX1, a cell-based screening assay was established. HEK293T cells were stably transfected with MRGPRX1 and stimulated with CQ. As shown in Figure 1A,B, CQ strongly increased intracellular calcium levels in a dose-dependent manner with an EC_50_ value of 96 μM in HEK-MrgprX1 cells. To identify MrgprX1 antagonists, HEK-MrgprX1 cells were loaded with Fluo-4 AM calcium indicator and treated with 30 μM of test compounds for 10 min, followed by stimulation with CQ. The screening of 2543 compounds revealed two antagonists of MrgprX1, berbamine and closantel, which inhibited >90% of the CQ-induced activation of MrgprX1 (Figure 1C,D).

### 2.2. Effect of Berbamine and Closantel on CQ- and Ionomycin-Induced Increases in Intracellular Calcium Levels

To investigate whether berbamine and closantel affect calcium signaling pathways, we observed the effects of berbamine and closantel on CQ- and ionomycin-induced increases in intracellular calcium concentrations in HEK-MrgprX1 and HEK293T cells, respectively. Berbamine and closantel both potently inhibited CQ-induced MrgprX1 activation in a dose-dependent manner with IC_50_ values of 1.6 μM and 3.6 μM, respectively (Figure 2A–C). However, in the case of the ionomycin-induced increase in intracellular calcium levels, berbamine had no effect, whereas closantel inhibited it in a dose-dependent manner (Figure 2D–F). These results indicated that berbamine inhibited the CQ-induced increase in intracellular calcium levels via MrgprX1 inhibition, so it was decided to conduct further studies on berbamine.

### 2.3. Berbamine Has No Effect on Other Pruritogenic GPCRs except for MrgprX1

Previous antagonists of MrgprX1 and MrgprA3 lack selectivity for other pruritogens [14,15]. To investigate the effect of berbamine on other pruritogenic GPCRs, we observed the effect of berbamine on histamine H_1_ receptor (H1R), histamine H_4_ receptor (H4R), MrgprC11, MrgprX4, PAR2, and TGR5. As shown in Figure 3, berbamine did not affect the histamine-induced activation of H1R and H4R, but the histamine-induced activation of H1R and H4R were significantly blocked by alcaftadine and JNJ-7777120, respectively. In the case of PAR2 and MrgprC11, berbamine did not affect the PAR2-activating peptide (AP)-induced activation of both PAR2 and MrgprC11. The PAR2-AP induced activation of PAR2 was significantly blocked by punicalagin, a potent antagonist of PAR2 [16]. In addition, berbamine did not alter the deoxycholic acid (DCA)-induced activation of both MrgprX4 and TGR5. These results showed that a high concentration (30 μM) of berbamine did not alter any of the agonist-induced activation of H1R, H4R, MrgprC11, MrgprX4, PAR2, or TGR5.

### 2.4. Berbamine Inhibits CQ-Induced ERK1/2 Phosphorylation through MrgprX1

MrgprX1 activation increased the phosphorylation of extracellular signal-regulated kinase (ERK) 1/2 in HEK293T cells expressing MrgprX1 [17]. Thus, we investigated whether CQ induced the phosphorylation of ERK1/2 in HEK-MrgprX1 cells. Strong and significant ERK1/2 phosphorylation was observed following CQ application, and ERK1/2 phosphorylation peaked 10 min after CQ treatment (Figure 4A,B). Interestingly, berbamine significantly reduced CQ-induced ERK1/2 phosphorylation (Figure 4C,D).

### 2.5. Berbamine Potently Inhibits MrgprA3-Mediated Calcium Signaling and ERK1/2 Phosphorylation

MRGPRA3 and MRGPRC11 are mouse orthologs of human MRGPRX1, and MrgprA3 is also activated by CQ, resulting in CQ-induced pruritus [7]. To investigate the inhibitory effect of berbamine on MrgprA3, we established HEK293T cells overexpressing MrgprA3. MrgprA3 was activated by CQ in a dose-dependent manner with an EC_50_ value of 234 μM (Figure 5A,B). Berbamine potently blocked CQ-induced MrgprA3 activation in a dose-dependent manner with an IC_50_ of 2.9 μM (Figure 5C,D). In the case of MrgprC11, berbamine showed minimal inhibitory effect on the activation of MrgprC11 by BAM8-22 at concentrations up to 30 μM (Figure 5E). We further investigated whether berbamine inhibited CQ-induced ERK1/2 phosphorylation in HEK-MrgprA3 cells. Strong and significant ERK1/2 phosphorylation was observed following CQ application, and ERK1/2 phosphorylation peaked 5 min after CQ treatment (Figure 6A,B). Interestingly, berbamine also significantly reduced CQ-induced ERK1/2 phosphorylation through MrgprA3 (Figure 6C,D). These results suggest that berbamine potently inhibits both MrgprX1 and MrgprA3 and is applicable to animal studies of CQ-induced pruritus.

### 2.6. Antipruritic Effect of Berbamine on CQ-Induced Pruritus Mouse Model

To investigate the antipruritic effect of berbamine, we observed the effect of berbamine on pruritogen-induced scratching behavior in BALB/c mice. Mice were transferred to clear cages and treated with berbamine intraperitoneally after a 30 min acclimatization period. Mouse behavior was recorded for 30 min right after the subcutaneous injection of pruritogens to the nape 30 min after berbamine treatment (Figure 7A). Interestingly, total scratching bouts due to the subcutaneous injection of CQ significantly were decreased by berbamine in a dose-dependent manner, attenuating 80% of CQ-induced pruritus at a dose of 30 mg/kg (Figure 7B). Notably, berbamine at a dose of 30 mg/kg significantly decreased the number of scratching bouts in all the intervals (Figure 7C). We further investigated the effect of berbamine on pruritus caused by other pruritogens. Interestingly, berbamine at a dose of 30 mg/kg showed no significant effect on total scratching bouts induced by histamine, PAR2-AP, and DCA (Figure 7D–F).

## 3. Discussion

The incidence of malaria increased from 56 in 2019 to 59 cases per 1000 population at risk in 2020 [1], and the emergence of resistant strains against the antimalarial drug has become a serious obstacle in the eradication and control of malaria [18]. Due to the lack of understanding of the CQ-induced pruritus and absence of therapeutic agents, the side effect of CQ may have contributed to the spread of CQ-resistant *Plasmodium falciparum*. The plasma concentration of CQ in patients is in the micromolar range, but due to its slow excretion and strong binding to melanin, CQ accumulates at considerable levels in pigmented tissue and skin, reaching high micromolar to millimolar concentrations [19,20,21,22,23,24,25]. Moreover, patients susceptible to CQ-induced pruritus are reported to reach higher concentrations of CQ in their skin than those not susceptible to the adverse effect [23].

Recently, a study revealed that human MrgprX1 and mouse MrgprA3 play pivotal roles in CQ-induced pruritus using Mrgpr-cluster△^−^/^−^ mice [7]. Interestingly, the Mrgpr-cluster△^−^/^−^ mice showed strong reduction in CQ-mediated pruritus while exhibiting normal behavior on acute pain and histamine-induced itching. Moreover gain- and loss-of-function research on utilizing neurons in Mrgpr-cluster△^−^/^−^ DRG strongly suggests the potential of MrgprX1 as a therapeutic target to modulate CQ-induced pruritus [7]. In the present study, we performed a cell-based screening using approved drugs and natural products to identify novel MrgprX1 antagonists and discovered two novel antagonists of MrgprX1, berbamine and closantel. Interestingly, berbamine had no effect on ionomycin-induced intracellular calcium signaling, whereas closantel dose dependently suppressed ionomycin-induced intracellular calcium levels (Figure 2D–F). In addition, berbamine did not activate MrgprX1 by itself (Appendix A), did not potently inhibit MrgprX1 activation by BAM8-22 (Appendix A), and exhibited a competitive inhibitory effect on the activation of MrgprX1 by CQ (Appendix A). Moreover, berbamine did not alter the other pruritogenic GPCRs, including H1R, H4R, MrgprC11, MrgprX4, PAR2, and TGR5 (Figure 3). These results suggest that berbamine is a bona fide MrgprX1 antagonist.

ERK regulates synaptic plasticity and central sensitization and is activated by acute and chronic noxious stimuli, thereby playing a crucial role in the development of acute and chronic pain and itch [26,27,28,29,30,31]. Recently, BAM8-22, an endogenous itch-inducing peptide that is a potent MrgprX1 agonist, was reported to induce EKR1/2 signaling at the cellular level using primary sensory neuron-derived F11 cells [17]. Here, we investigated whether CQ could induce ERK1/2 phosphorylation through MrgprX1 and found that CQ significantly activated ERK1/2 signaling, peaking at 10 min (Figure 4A,B). Moreover, berbamine strongly suppressed CQ-induced phosphorylation in a dose-dependent manner (Figure 4C,D). These results suggest that the CQ-induced phosphorylation of ERK1/2 via MrgprX1 may play an important role in CQ-induced pruritus and that berbamine can attenuate CQ-induced pruritus through MrgprX1 inhibition.

Human MRGPRX1 shares homology with mouse MRGPRC11 and MRGPRA3. However, human MrgprX1 can be activated by both BAM8-22 and CQ, whereas mouse MrgprC11 is only activated by BAM8-22, and mouse MrgprA3 is only activated by CQ [7]. Here, we investigated whether berbamine could also inhibit MrgprA3-mediated calcium signaling and ERK1/2 phosphorylation by CQ and found that berbamine potently blocked CQ-induced calcium signaling (Figure 5A–D) and significantly decreased the phosphorylation of ERK1/2 (Figure 6). Notably, berbamine exhibited a minimal inhibitory effect on the activation of MrgprC11 by BAM8-22 (Figure 5E). These results suggest that berbamine is a suitable MrgprA3 antagonist for CQ-induced pruritus mouse models. In this study, we did not elucidate the inhibitory effect of berbamine on CQ-induced calcium signaling in small-diameter dorsal root ganglion (DRG) neurons. Further studies demonstrating the inhibitory effects of berbamine in small-diameter DRG neurons could provide useful information for the development of new therapeutics for CQ-induced pruritus.

In our pruritogen-induced pruritus mouse model, berbamine significantly inhibited CQ-induced pruritus in a dose-dependent manner but did not alter histamine-, PAR2-AP-, or DCA-induced itch (Figure 7). Interestingly, Mrgpr-cluster△^−^/^−^ mice showed approximately 65% reduction in CQ-induced scratching [7], whereas 30 mg/kg berbamine reduced CQ-induced scratching by ~80% (Figure 7). The stronger inhibitory effect of berbamine compared with Mrgpr-cluster△^−^/^−^ mice may be due to differences in mouse strains or off-target effects of berbamine.

Previous studies have shown that crotamiton inhibits not only CQ-induced itch but also histamine-induced itch; the effects of JHU739 on other pruritogenic GPCRs are not known, and further studies are required [14,15,32]. In this study, we showed that berbamine did not affect other pruritogenic GPCRs except for MrgprX1 (Figure 3). These results suggest that berbamine can be utilized as a useful pharmacological tool to dissect MrgprX1 from other pruritogenic GPCRs. In addition, in terms of drug development, the non-specific inhibition of histamine receptors can lead to common side effects, such as drowsiness, dizziness, dry mouth, and constipation. Thus, the selective inhibition of MrgprX1 by berbamine avoids the side effects caused by the inhibition of other pruritic GPCRs, including histamine receptors. Moreover, berbamine was reported to have a long half-life (39.25 h) in a human pharmacokinetic study [33].

Interestingly, a previous study showed that berbamine also has antimalaria activity itself in vitro [34]. Berbamine showed antimalaria effect with IC_50_ values of 603 ± 47 nM and 359 ± 9 nM against CQ-sensitive and -resistant strains of *P. falciparum*, respectively. In the case of the CQ-resistant strain, the IC_50_ value of CQ was greatly reduced when combined with berbamine. For example, the IC_50_ of CQ against a CQ-resistant strain was 203.2 ± 70.7 nM, but it was decreased to 26.3 ± 4.5 nM when given together with 1 μM berbamine [34]. Taken together, these results suggest that berbamine not only attenuates CQ-induced pruritus but also has the potential to significantly increase the antimalarial therapeutic effect against CQ-resistant strains.

## 4. Materials and Methods

### 4.1. Cell Culture

HEK293T and CHO-K1 were cultured in Dulbecco’s modified Eagle medium (DMEM). DMEM was supplemented with 10% fetal bovine serum (FBS), 100 U/mL penicillin, and 100 μg/mL streptomycin. HT29 were cultured in Roswell Park Memorial Institute (RPMI) 1640 supplemented with 10% fetal bovine serum (FBS), 100 U/mL penicillin, and 100 μg/mL streptomycin. All cells were grown at 37 °C, 5% CO_2_, and 95% humidity.

### 4.2. Intracellular Calcium Measurement

Intracellular calcium levels were measured using a Fluo-4 NW calcium assay kit (Invitrogen, Carlsbad, CA) as per the manufacturer’s protocol in HEK293T and HT29 cells. Briefly, the cells were plated in a 96-well black-walled plate and incubated with 100 μL of assay buffer including Fluo-4 NW. After 1 h of incubation, Fluo-4 fluorescence was recorded with a FLUOstar Omega microplate reader equipped with syringe pumps and custom Fluo-4 excitation/emission filters (485/538 nm). Intracellular calcium increase was induced with the application of the indicated agonist. All modulators were dissolved in DMSO, and cells were treated with a final concentration of 1% DMSO.

### 4.3. YFP Fluorescence Quenching Assay

TGR5-, CFTR-, and halide sensor YFP-F46L/H148Q/I152L-expressing CHO-K1 cells were plated in 96-well black-walled microplates (Corning Inc., Corning, NY, USA) at a confluence of ~80% per well. Assays were performed using a FLUOstar Omega microplate reader (BMG Labtech, Ortenberg, Germany) and MARS Data Analysis Software (BMG Labtech) as described in our previous study [35]. Briefly, each well of the 96-well plates was washed 3 times in PBS (200 μL/wash), and 80 μM DCA and DMSO or 30 μM berbamine was applied to each well. After 10 min of incubation at 37 °C, the 96-well plates were transferred into the plate reader for the fluorescence quenching assay. Each well was measured individually for CFTR^-^-mediated I^-^ influx by recording YFP fluorescence continuously (400 ms per point) for 1 s (baseline); then, 140 mM I^-^ solution was injected at 1 s, and YFP fluorescence was recorded for 7 s. The initial iodide influx rate was determined from the initial slope of fluorescence decrease, using nonlinear regression, following the infusion of iodide. All modulators were dissolved in DMSO, and cells were treated with a final concentration of 1% DMSO.

### 4.4. Immunoblot Analysis

For Western blot analyses, cells were plated on 6-well plates and were serum-starved overnight. Cells were treated with compounds accordingly, and protein samples were prepared as described previously [36]. Then, 30 µg of total proteins was loaded to each well, and proteins were separated using 4–12% Tris Glycine Precast Gel (KOMA BIOTECH, Seoul, Republic of Korea). Separated proteins were transferred to polyvinylidene fluoride (PVDF) membranes. Blocking was carried out using Tris-buffered saline with 0.1% Tween 20 (TBST) containing 5% BSA at room temperature for 1 h. Then, the membranes were incubated with primary antibodies overnight at 4 °C with the indicated antibodies: anti-p42/44 (Cell Signaling; Cat#9102S) and anti-phospho-p42/44 (Cell Signaling; Cat#9101L). Subsequently, the membranes were washed out with TBST 3 times at 5 min intervals and incubated with HRP-conjugated anti-secondaries for 1 h at room temperature. Finally, visualization was carried out using an ECL Plus immunoblotting detection system (GE Healthcare, Piscataway, NJ, USA). All experiments were repeated five times independently, and ImageJ software (NIH, Bethesda, MD, USA) was used for analysis.

### 4.5. Acute Pruritogen-Induced Pruritus

Control-region-imprinted six-week-old BALB/c male mice were purchased from DBL (Chungcheong-do, Republic of Korea), and an additional 1 week was given for accommodation. Berbamine was administered intraperitoneally after 30 min of acclimatization. After 30 min, pruritogen compounds (i.e., CQ, histamine, PAR2-AP, and DCA) were subcutaneously administered into the nape, and scratching bouts were recorded for 30 min. Continuous scratch movements with hind limbs were defined as a bout of scratching. The total number of scratching bouts was counted by recording the number of scratching bouts in the recorded video over 30 min of observation time.

### 4.6. Statistical Analysis

All experiments were performed independently a minimum of five times. Statistical analyses were performed using GraphPad Prism 9.0 (GraphPad Software Inc., San Diego, CA, USA). The results for multiple trials are presented as the mean ± standard deviation (S.D.) or mean ± standard error (S.E.). One-way ANOVAs with Tukey’s post hoc tests and Student’s *t*-tests were used to conduct the statistical analyses. Statistical significance was considered at *p* values < 0.05.

### 4.7. Study Approval

All procedures of animal experiments were performed by Yonsei University Animal Care and Use Committee (Seoul, Republic of Korea).

## Figures and Tables

**Figure 1 ijms-23-14321-f001:**
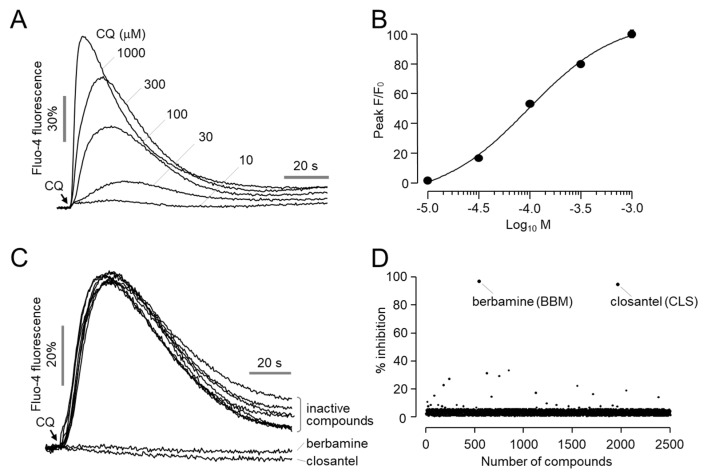
Identification of novel antagonists of MrgprX1: (**A**) Intracellular calcium levels were measured using Fluo−4 NW in HEK293T cells expressing human MrgprX1. Cells were treated with the indicated concentrations of CQ where the arrow indicates. (**B**) Summary of dose-response (mean ± S.E., *n* = 6). (**C**) Representative Fluo−4 fluorescence traces of MrgprX1 antagonist screening. Cells were treated with 30 μM of the test compounds for 10 min prior to treatment with 300 μM CQ. (**D**) Scatter plot showing the results of MrgprX1 antagonist screening for 2543 compounds.

**Figure 2 ijms-23-14321-f002:**
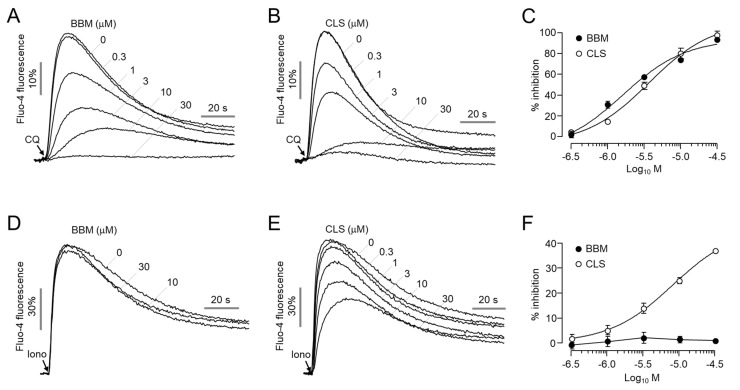
Effect of berbamine and closantel on CQ- and ionomycin-induced increases in intracellular calcium levels: (**A**,**B**) Intracellular calcium levels were measured using Fluo−4 NW in HEK293T cells expressing MrgprX1. The cells were pretreated with the indicated concentrations of berbamine (BBM) and closantel (CLS) for 10 min prior to treatment with 300 μM CQ. (**C**) Summary of dose-response (mean ± S.E., *n* = 6). (**D**,**E**) Intracellular calcium levels were increased with 10 μM ionomycin in HEK239T cells. The cells were pretreated with the indicated concentrations of BBM and CLS for 10 min before treatment with ionomycin. (**F**) Summary of dose–response (mean ± S.E., *n* = 6).

**Figure 3 ijms-23-14321-f003:**
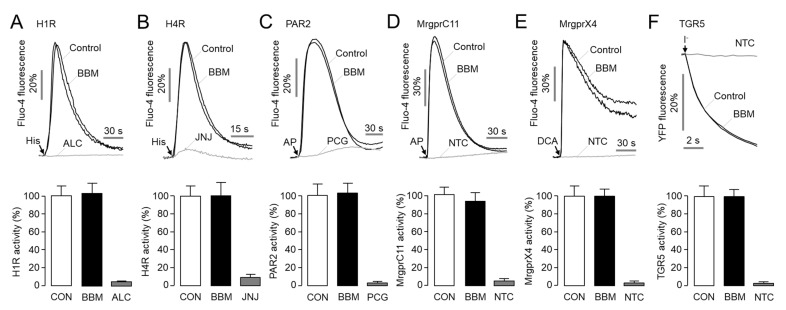
Effects of berbamine on the activity of other pruritogenic GPCRs: (**A**,**B**) Effect of berbamine on histamine 1 receptor (H1R) and histamine 4 receptor (H4R). H1R- and H4R-expressing HEK293T cells were pretreated with 30 μM BBM for 10 min prior to 100 μM histamine treatment. H1R and H4R were inhibited by 30 μM alcaftadine (ALC) and JNJ-7777120 (JNJ), respectively. (**C**) Effect of berbamine on PAR2. HT29 cells were pretreated with BBM for 10 min prior to 30 μM PAR2-AP treatment. PAR2 was inhibited by 30 μM punicalagin (PCG). (**D**) Effect of berbamine on MrgprC11. MrgprC11-expressing HEK293T cells were pretreated with BBM for 10 min prior to 100 μM PAR2-AP treatment. (**E**) Effect of berbamine on MrgprX4. MrgprX4-expressing HEK293T cells were pretreated with BBM for 10 min prior to 80 μM DCA treatment. (**F**) Effect of berbamine on TGR5. TGR5 activity was measured using a YFP fluorescence quenching assay. CHO-K1 cells coexpressing TGR5 and a halide sensor YFP were pretreated with BBM and 80 μM DCA for 10 min prior to I^-^ treatment. Results were summarized as means ± S.E. (*n* = 6). NTC, nontransfected cells.

**Figure 4 ijms-23-14321-f004:**
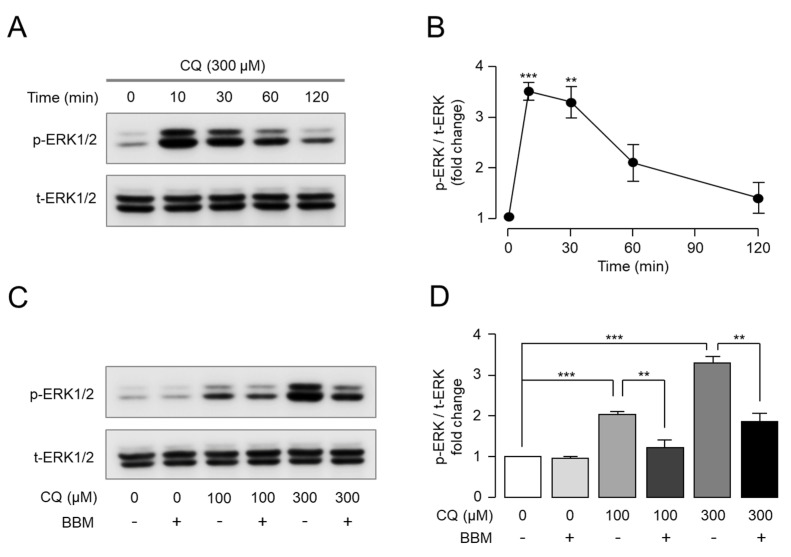
Effect of berbamine on CQ-induced ERK1/2 phosphorylation through MrgprX1: (**A**) Representative immunoblot analysis results of ERK1/2 phosphorylation in CQ-treated HEK293T cells expressing MrgprX1. (**B**) Phosphorylated ERK1/2 was normalized to total ERK1/2 (mean *±* S.E., *n* = 5). Statistical significance of differences between each time point and 0 min was determined using Student’s *t*-test. ** *p* < 0.01, *** *p* < 0.001. (**C**) Effect of BBM on CQ-induced phosphorylation of ERK1/2. Cells were pretreated with 10 μM BBM for 10 min prior to CQ treatment. (**D**) Phosphorylated ERK1/2 was normalized to total ERK1/2 (mean *±* S.E., *n* = 5). Statistical significance of differences between indicated groups was assessed using one-way ANOVA with Tukey’s post hoc test. ** *p* < 0.01, *** *p* < 0.001.

**Figure 5 ijms-23-14321-f005:**
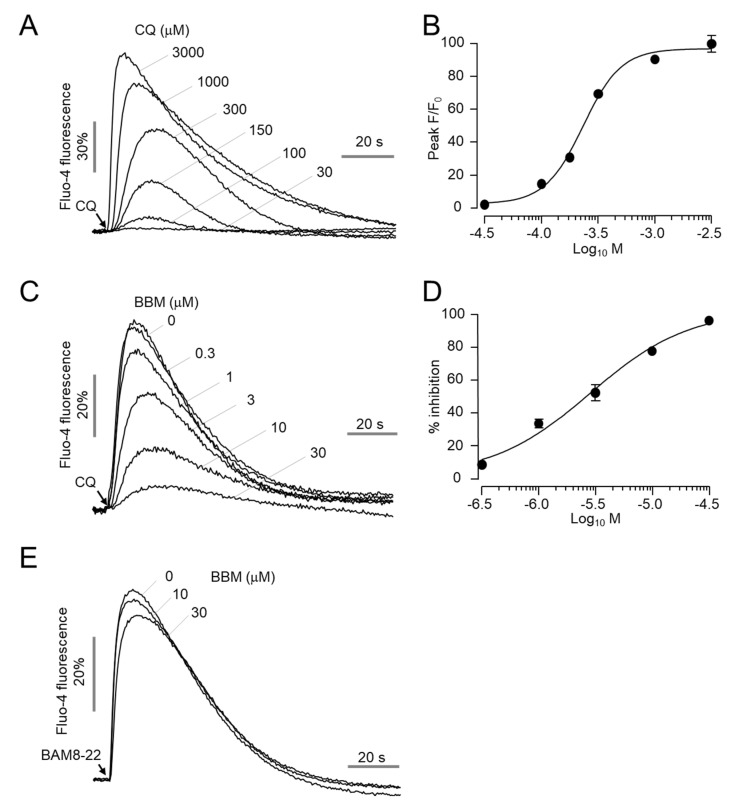
Effect of berbamine on CQ-induced activation of MrgprA3: (**A**) Intracellular calcium levels were measured in HEK293T cells expressing mouse MrgprA3. Cells were treated with indicated concentrations of CQ. (**B**) Summary of dose–response (mean ± S.E., *n* = 6). (**C**) Cells were pretreated with indicated concentrations of berbamine for 10 min prior to 300 μM CQ treatment. (**D**) Summary of dose-response (mean ± S.E., *n* = 6). (**E**) Representative traces of intracellular calcium responses to BAM8−22 in HEK293T cells expressing MrgprC11. Cells were pretreated with the indicated concentrations of berbamine for 10 min before treatment with 500 nM BAM8−22.

**Figure 6 ijms-23-14321-f006:**
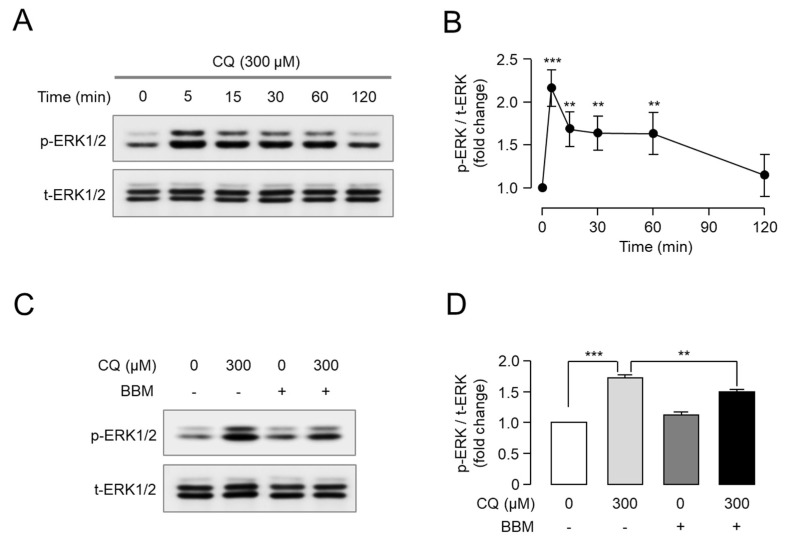
Effect of berbamine on CQ-induced ERK1/2 phosphorylation through MrgprA3: (**A**) Representative immunoblot analysis results of ERK1/2 phosphorylation in CQ-treated HEK293T cells expressing MrgprA3. (**B**) Phosphorylated ERK1/2 was normalized to total ERK1/2 (mean *±* S.E., *n* = 5). Statistical significance of differences between each time point and 0 min was determined using Student’s *t*-test. ** *p* < 0.01, *** *p* < 0.001. (**C**) Cells were pretreated with 30 μM BBM for 10 min prior to CQ treatment. (**D**) Phosphorylated ERK1/2 was normalized to total ERK1/2 (mean *±* S.E., *n* = 3). Statistical significance of differences between indicated groups was determined using Student’s *t*-test. *** *p* < 0.001.

**Figure 7 ijms-23-14321-f007:**
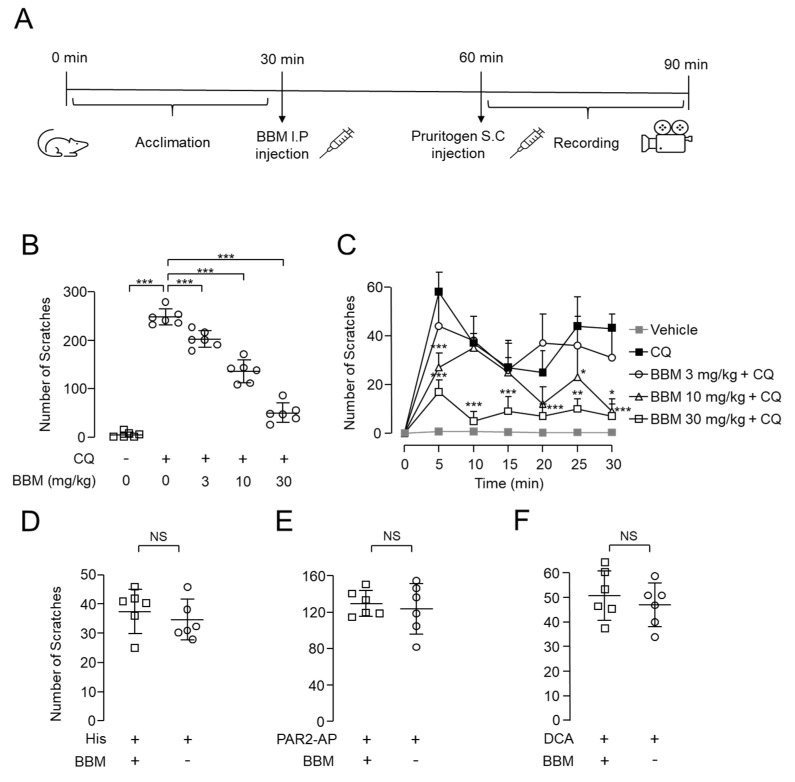
Effect of berbamine on acute pruritogen-induced pruritus in mice: (**A**) Scheme of acute pruritogen-induced pruritus mouse model. (**B**) Effect of BBM on scratching bouts during 30 min after subcutaneous injection of CQ (400 µg/50 µL). (**C**) Time-course graphs of scratching bouts at 5 min intervals (mean ± S.D., *n* = 6). Statistical significance of differences between each group and CQ group was assessed using one-way ANOVA with Tukey’s post hoc test. * *p* < 0.05, ** *p* < 0.01, *** *p* < 0.001. (**D**–**F**) Effect of BBM on the scratching bouts during 30 min after subcutaneous injection of histamine (10 μM), PAR2-AP (100 μg/50 μL), and DCA (50 μg/50 μL). Results were summarized as mean ± S.D. (*n* = 6). Statistical significance of differences was determined using Student’s *t*-test. NS, not significant (*p* > 0.05).

## Data Availability

Not applicable.

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
