# Peer review of "Berbamine Reduces Chloroquine-Induced Itch in Mice through Inhibition of MrgprX1"

_ijms, 2022, doi:10.3390/ijms232214321_

Round 1

Reviewer 1 Report

The authors report that the chemical berbamine acts as an antagonist at MRGPRX1 receptors, which are expressed by a subset of peripheral neurons that specifically transmit itch signals. Highly specific antagonists for MRGPRX1 not widely available to the scientific community, so this study is potentially of great value for the field. The report is cleanly written but lacks some critical details on berbamine’s actions.

Major Concerns

1. MRGPRX1 can be activated by CQ and BAM8-22, but the authors only test CQ. They should test the effects of berbamine on BAM8-22 activation, as well, at MRGPRX1 and Mrgprc11 receptors.  

2. The authors should test whether berbamine activates MRGPRX2 or Mrgprb2. This is critical because these are expressed by mast cells, and if berbamine activates them, it could cause serious side effects.

3. The authors should report whether berbamine triggers any activation of MRGPRX1 on its own.

4. It is not clear how berbamine works; for instance, whether it is a competitive or non-competitive antagonist. The authors should try using a fixed concentration of berbamine (perhaps 30 micromolar) and different CQ concentrations to see if the dose-response curve is simply shifted to the right (competitive antagonist), or if the peak response to CQ is altered (more likely non-competitive).

5. In Figure 3, the authors use fixed concentrations of receptor agonists and berbamine. The agonist concentrations often are quite high and may overwhelm any antagonist effect of berbamine. The authors should try at least one lower drug concentration to see if an effect is seen at more moderate receptor stimulation conditions.

6. In Figure 6, the authors report a modest effect on ERK phosphorylation by berbamine, though this concentration nearly abolishes calcium mobilization. Can the authors comment on the different effects?

7. The authors should do a scratching assay to see if berbamine itself can trigger scratching.

8. In Figure 7b, the authors show that berbamine can almost fully block CQ-induced scratching. However, in the Cell paper that introduced CQ as an Mrgpra3 agonist, knocking out Mrgpra3 only reduced scratching by about 50%. This implies that berbamine has off-target effects. Can the authors provide a commentary on why this is observed, and the potential implications?

Minor Concerns

1. The authors don’t appear to list how berbamine is dissolved in their assay solutions. It looks like it’s dissolved in DMSO, but the stock concentration and final DMSO concentrations are not listed. This is required for others to be able to confirm their results.

2. In the introduction, line 44 on the first page, the authors claim that “Mrgprs are distinctively expressed in small diameter dorsal root ganglia (DRG) neurons and trigeminal ganglia neurons…”. This is only true of some of the MRGPR family and most definitely is not true for all members. For instance, MRGPRX2 is expressed by mast cells, and MRGPRF is quite widely expressed in non-neuronal cells.

3. The authors should include more information about berbamine in the discussion to help the reader evaluate whether it’s suitable for in vivo use. Specifically, any data on its distribution in the body, half life, usual dosage, and toxicity should be included.

Author Response

We greatly appreciate the editor’s and reviewers’ efforts to carefully review our manuscript and the valuable comments and suggestions offered for the improvement of the manuscript (ijms-1997952). We have made each of the suggested revisions. The points of criticism raised by the reviewers were addressed by a point-by-point response. Changes in the manuscript text are highlighted in red color font.

Reviewer #1:

The authors report that the chemical berbamine acts as an antagonist at MRGPRX1 receptors, which are expressed by a subset of peripheral neurons that specifically transmit itch signals. Highly specific antagonists for MRGPRX1 not widely available to the scientific community, so this study is potentially of great value for the field. The report is cleanly written but lacks some critical details on berbamine’s actions.

Major Concerns

  1. MRGPRX1 can be activated by CQ and BAM8-22, but the authors only test CQ. They should test the effects of berbamine on BAM8-22 activation, as well, at MRGPRX1 and MrgprC11 receptors.

Response: Thank you for the helpful suggestions. We conducted additional experiments to observe the effect of berbamine on the activation of MrgprX1 and MrgprC11 by BAM8-22. As shown Figure S2, berbamine inhibited MrgprX1 activation by BAM8-22 much less potently than MrgprX1 activation by CQ. Interestingly, berbamine showed very weak inhibitory effect on the activation of MrgprC11 by BAM8-22 (Figure 5E). These results suggest that berbamine has a stronger inhibitory effect on CQ-induced MrgprX1 and MrgprA3 activation than BAM8-22-induced MrgprX1 and MrgprC11 activation. The above is described in the revised manuscript.

  1. The authors should test whether berbamine activates MRGPRX2 or Mrgprb2. This is critical because these are expressed by mast cells, and if berbamine activates them, it could cause serious side effects.

Response: Thank you for the comment. To investigate whether berbamine activates MRGPRX2, we observed the degranulation effect of berbamine in rat basophilic lukemia-2H3 (RBL-2H3) overexpressing MRGPRX2 mast cells. Berbamine did not induce degranulation in RBL-2H3 mast cells as shown below (please find the result in the rebuttal letter file). This result suggests that berbamine does not activate MrgprX2.

  1. The authors should report whether berbamine triggers any activation of MRGPRX1 on its own.

Response: Thank you for the helpful comment. We observed whether berbamine triggers activation of MRGPRX1. As shown in Figure S1, berbamine did not activate MrgprX1 on its own.

  1. It is not clear how berbamine works; for instance, whether it is a competitive or non-competitive antagonist. The authors should try using a fixed concentration of berbamine (perhaps 30 micromolar) and different CQ concentrations to see if the dose-response curve is simply shifted to the right (competitive antagonist), or if the peak response to CQ is altered (more likely non-competitive).

Response: Thank you for the helpful suggestion. Pretreatment of berbamine (30 mM) shifted dose-response curve of CQ to the right suggesting berbamine as a competitive antagonist (Figure S3). Unfortunately, full dose-response studies could not be performed due to solubility issues of CQ.

  1. In Figure 3, the authors use fixed concentrations of receptor agonists and berbamine. The agonist concentrations often are quite high and may overwhelm any antagonist effect of berbamine. The authors should try at least one lower drug concentration to see if an effect is seen at more moderate receptor stimulation conditions.

Response: Thank you for the comment. In Figure 3, we observed the effects of berbamine at a concentration capable of completely inhibiting MrgprX1 on other pruritogenic GPCRs. Agonists for each GPCR were used at concentrations that were almost completely blocked by selective antagonists for each GPCR. So, we believe that the concentration of agonists did not overwhelm the antagonistic effect of berbamine.

  1. In Figure 6, the authors report a modest effect on ERK phosphorylation by berbamine, though this concentration nearly abolishes calcium mobilization. Can the authors comment on the different effects?

Response: Thank you for the helpful comment. In this study, we could not clarify the different effect between ERK phosphorylation and calcium mobilization. However, we understand that it is a phenomenon that occurs because ERK phosphorylation is affected by more complex pathways than calcium mobilization in general.

  1. The authors should do a scratching assay to see if berbamine itself can trigger scratching.

Response: Thank you for the helpful suggestion. No itching behavior was observed for 30 minutes when berbamine was administered intraperitoneally, so it seems that there is no scratching-inducing effect of berbamine itself.

  1. In Figure 7b, the authors show that berbamine can almost fully block CQ-induced scratching. However, in the Cell paper that introduced CQ as an Mrgpra3 agonist, knocking out Mrgpra3 only reduced scratching by about 50%. This implies that berbamine has off-target effects. Can the authors provide a commentary on why this is observed, and the potential implications?

Response: Thank you for the helpful comments. MrgprA3 knock out mouse showed approximately 65% of reduction in scratching (Qin et al., Cell, 2009). However, berbamine reduced roughly 80% of CQ-induced scratching. This difference may be due to off-target effects of berbamine, as the reviewer comments, or due to differences in mouse strains. This phenomenon is described in the revised discussion section.

Minor Concerns

  1. The authors don’t appear to list how berbamine is dissolved in their assay solutions. It looks like it’s dissolved in DMSO, but the stock concentration and final DMSO concentrations are not listed. This is required for others to be able to confirm their results.

Response: Thank you for the comment. The information is described in the revised Method section.

  1. In the introduction, line 44 on the first page, the authors claim that “Mrgprs are distinctively expressed in small diameter dorsal root ganglia (DRG) neurons and trigeminal ganglia neurons…”. This is only true of some of the MRGPR family and most definitely is not true for all members. For instance, MRGPRX2 is expressed by mast cells, and MRGPRF is quite widely expressed in non-neuronal cells.

Response: We agree with the reviewer's comment. The sentence has been clearly corrected.

  1. The authors should include more information about berbamine in the discussion to help the reader evaluate whether it’s suitable for in vivo use. Specifically, any data on its distribution in the body, half life, usual dosage, and toxicity should be included.

Response: Thank you for the comment. More details about berbamine are provided in the revised Discussion section.

Reviewer 2 Report

Chloroquine is a widely used drug to treat malaria, but its use can lead to severe itch in certain patients.  Although MrgprX1 has been identified as the primary receptor mechanism underlying chloroquine induced itch, treatments are still lacking.  Ryu, et al. present findings identifying the compound Berbamine as a selective antagonist for MrgprX1.  The work by Ryu and colleagues is well put together and detailed, with extensive findings and appropriate controls. 

My only major concern is the lack of any testing of the functional consequence of Berbamine on chloroquine activation of small diameter dorsal root ganglia (DRG) neurons.  The calcium imaging studies are confined to expression systems (HEK and CHO cells).  A small calcium imaging experiment whereby pre-application of Berbamine can inhibit activation of DRG neurons would greatly strengthen the results.  

Author Response

We greatly appreciate the editor’s and reviewers’ efforts to carefully review our manuscript and the valuable comments and suggestions offered for the improvement of the manuscript (ijms-1997952). We have made each of the suggested revisions. The points of criticism raised by the reviewers were addressed by a point-by-point response. Changes in the manuscript text are highlighted in red color font.

Reviewer #2:

Chloroquine is a widely used drug to treat malaria, but its use can lead to severe itch in certain patients.  Although MrgprX1 has been identified as the primary receptor mechanism underlying chloroquine induced itch, treatments are still lacking.  Ryu, et al. present findings identifying the compound Berbamine as a selective antagonist for MrgprX1. The work by Ryu and colleagues is well put together and detailed, with extensive findings and appropriate controls.

My only major concern is the lack of any testing of the functional consequence of Berbamine on chloroquine activation of small diameter dorsal root ganglia (DRG) neurons.  The calcium imaging studies are confined to expression systems (HEK and CHO cells). A small calcium imaging experiment whereby pre-application of Berbamine can inhibit activation of DRG neurons would greatly strengthen the results. 

Response: Thank you for the thoughtful comments. We agree with the reviewer’s opinion, but unfortunately, we could not investigate the inhibitory effect of berbamine in DRG neurons in this study due to several limitations. This issue was addressed in the discussion section of the revised manuscript.

Round 2

Reviewer 1 Report

I appreciate all of the work the authors have performed, and they have satisfied the concerns I had listed. However, I now have one additional recommendation, that the authors should examine antagonism at MRGPRX2 and Mrgprb2 receptors (I previously had recommended checking for agonism, not antagonism). I apologize for this, but it is extremely important because mast cell degranulation by these receptors also produces strong itch, and it's critical to rule out effects on these receptors when using berbamine in itch studies.

Author Response

Reviewer #1:

I appreciate all of the work the authors have performed, and they have satisfied the concerns I had listed. However, I now have one additional recommendation, that the authors should examine antagonism at MRGPRX2 and Mrgprb2 receptors (I previously had recommended checking for agonism, not antagonism). I apologize for this, but it is extremely important because mast cell degranulation by these receptors also produces strong itch, and it's critical to rule out effects on these receptors when using berbamine in itch studies.

Response: Thank you for the comment. Fortunately, we already have the results on the inhibitory effect of berbamine on MrgprX2 activation. To investigate whether berbamine inhibits MrgprX2 activity, we observed the degranulation effect of berbamine in rat basophilic lukemia-2H3 (RBL-2H3) overexpressing MRGPRX2 mast cells. Berbamine did not reduce Substance P-induced degranulation in RBL-2H3 cells as shown below (please find the result in rebuttal letter file). This result suggests that berbamine does not affect MrgprX2 activity.

Reviewer 2 Report

It is unclear to me in the text what the limitations were as to why the authors could not do a DRG calcium imaging experiment.

Author Response

Reviewer #2:

It is unclear to me in the text what the limitations were as to why the authors could not do a DRG calcium imaging experiment.

Response: Sorry for the vague response. Unfortunately, my lab currently does not have a setting capable of isolating DRG neurons. It takes several months to obtain animal research ethics permission and to set up isolating DRG neurons or conduct joint research. In addition, the first author should publish the manuscript as soon as possible to obtain his Ph.D degree. We did our best to do additional in vitro experiments and try to improve the quality of the manuscript. However, it was difficult to see the effect of berbamine on DRG neurons in a short time. We hope you understand this situation.